# Addressing the mean-correlation relationship in co-expression analysis

**Yi Wang**[1], **Stephanie C. Hicks**[1], **Kasper D. Hansen**[1,2]*

**1** Department of Biostatistics, Johns Hopkins Bloomberg School of Public Health, Baltimore, Maryland, United States of America, **2** McKusick-Nathans Department of Genetic Medicine, Johns Hopkins School of Medicine, Baltimore, Maryland, United States of America

* khansen@jhsph.edu

**Data Availability Statement:** The spqn package is available at http://www.bioconductor.org/packages/spqn. The script for the analysis is available at https://github.com/hansenlab/spqn_paper. The script for GTEx data processing is

## Abstract

Estimates of correlation between pairs of genes in co-expression analysis are commonly used to construct networks among genes using gene expression data. As previously noted, the distribution of such correlations depends on the observed expression level of the involved genes, which we refer to this as a *mean-correlation relationship* in RNA-seq data, both bulk and single-cell. This dependence introduces an unwanted technical bias in co-expression analysis whereby highly expressed genes are more likely to be highly correlated. Such a relationship is not observed in protein-protein interaction data, suggesting that it is not reflecting biology. Ignoring this bias can lead to missing potentially biologically relevant pairs of genes that are lowly expressed, such as transcription factors. To address this problem, we introduce spatial quantile normalization (SpQN), a method for normalizing local distributions in a correlation matrix. We show that spatial quantile normalization removes the mean-correlation relationship and corrects the expression bias in network reconstruction.

## Author summary

Coordinated changes in gene expression are usually interpreted as evidence of coordinated regulation or functional relatedness, in a type of analysis called co-expression analysis. It has previously been noted that genes with high expression level are more likely to exhibit coordinated expression with other genes, and that this causes a bias in co-expression analysis. Here, we study this bias and develop a method to correct it. After applying our method, which we call spatial normalization (SpQN), there is no longer a dependency between expression level and expression coordination.

This is a *PLOS Computational Biology* Methods paper.

## Introduction

Gene co-expression analysis is the study of correlation patterns in gene expression data, usually with the goal of constructing gene networks [1, 2]. Amongst popular methods for gene co-

available in the spqnData package at http://bioconductor.org/packages/spqnData. The processed datasets are available at https://zenodo.org/record/5784626#.YcC7lr3MLxR. The GTEx datasets used in this study were downloaded from from GTEx portal https://gtexportal.org/home/datasets. The drosophila bulk RNA-seq dataset and metadata were downloaded from http://bowtie-bio.sourceforge.net/recount/pooled/modencodefly_pooledreps_count_table.txt and http://bowtie-bio.sourceforge.net/recount/pooled/modencodefly_pooled_phenodata.txt. The scRNA-seq dataset is available from GEO with accession number GSE45719. The protein-protein interaction database was downloaded from http://www.interactome-atlas.org/data/HuRI.tsv. The regulatome data was curated using the code from https://github.com/leekgroup/networks_correction/blob/master/shellscripts/get_true_positive.sh. The transcription factor gene list was downloaded from: https://www.ncbi.nlm.nih.gov/pmc/articles/PMC4825693/bin/NIHMS772895-supplement-Table_S2.xlsx.

**Funding:** Research reported in this publication was supported by the National Institute of General Medical Sciences of the National Institutes of Health under award number R01GM121459 (to KDH), the National Human Genome Research Institute of the National Institutes of Health under the award number R00HG009007 (to SCH) and CZF2019-002443 and CZF2018-183446 from the Chan Zuckerberg Initiative DAF, an advised fund of Silicon Valley Community Foundation. The funders had no role in study design, data collection and analysis, decision to publish, or preparation of the manuscript.

**Competing interests:** The authors have declared that no competing interests exist.

expression analysis is WGCNA [3] which works on the correlation matrix and the graphical LASSO [4] which works on the precision matrix, the inverse of the covariance matrix. These methods have been successfully used many times to gain biological insight [5–9].

While co-expression analysis is widely used, there has been less work on various sources of confounding and bias, especially in contrast to the rich literature on such issues in the related field of differential gene expression analysis. Recent work in co-expression analysis is addressing this shortcoming, including work on removing unwanted variation [10, 11] and the effect of cell type composition [12].

The relationship between gene expression level and co-expression signals was firstly explored in [13]. The authors show a relationship between co-expression and mean expression level, "making expression level itself highly predictive of network topology" [13]. The authors focus on single-cell RNA-seq data, but state that the relationship is also present in bulk RNA-seq, including in meta-analyses of multiple datasets. They show that controlling for gene expression level changes the interpretation of their data, but do not provide any insight into the origin of the relationship. The authors strongly recommend to control for gene expression level, and recommend doing it by matching on expression level. Such a strategy is easy to apply when the goal is to examine the co-expression level of a fixed gene set, but it is unclear how such a strategy can be applied to constructing a complete gene network.

Recently, it was also shown that differential expression confounds differential co-expression analysis [14]. Defining differential co-expression as changes in correlation patterns between conditions, the authors show that most correlation changes are associated with changes in gene expression levels in the same genes between conditions, and provide a method to control for this confounding effect when identifying differential co-expression. The method cannot be used to control for the impact of expression level in network inference per se, only in the differential setting. This work highlights the importance of considering changes in expression level when interpreting changes in correlation patterns.

# Materials and methods

## GTEx bulk RNA-seq dataset

We used GTEx dataset for 9 tissues [15], including adipose subcutaneous, adrenal gland, artery tibial, brain cerebellum, brain cerebellum, brain cortex, breast mammary, colon transverse, esophagus mucosa and heart left ventricle. The read counts data were downloaded from GTEx portal, release v6p, expression measures `GTEx_Analysis_v6p_RNA-seq_RNA-SeQCv1.1.8_gene_reads.gct.gz`, gene annotation file `gencode.v19.genes.v6p_model.patched_contigs.gtf.gz` and sample information file `GTEx_Data_V6_Annotations_SampleAttributesDS.txt`. We kept protein coding genes and long non-coding genes, and transformed the read counts into log-RPKMs scale using

$$\log_2\left(\frac{\text{number of reads} + 0.5}{\text{library size} * \text{gene length}} * 10^9\right)$$

where library size is the total number of reads in the sample. For each tissue, we only kept the genes with median $\log_2(\text{RPKM})$ above 0.

## Single-cell RNA-seq dataset

We downloaded the scRNA-seq data from [16]. RPKM normalized expression matrix of 60 midblast cells are used. We transformed the expression matrix with $\log_2(\text{RPKM} + 0.5)$, and kept the genes with median $\log_2(\text{RPKM} + 0.5)$ greater than 0.

### Drosophila developmental time course

The bulk RNA-seq dataset of drosophila was downloaded from ReCount [17]. The dataset is composed of samples across 30 different developmental stage. The gene expression counts matrix of the same drophila were pooled into one sample in the analysis.

The dataset contains 14869 genes. The gene expression counts data was normalized using $\log_2(\text{RPM})$. Genes with median $\log_2(\text{RPM})$ above 0 were kept, and 9719 genes were kept. The expression level of each gene was scaled and top 5 principle components was removed.

### PPI database

We downloaded PPI data from HuRI database, which contains around 53,000 pairs of protein-protein interactions in human [18], with each protein annotated by the corresponding gene ensemble IDs. For each tissue, we only kept the genes overlapped with the genes in the filtered counts.

### Regulatome database

We used regulatome data collected in [11], which contains 2269 regulatory pathways, with 10198 genes involved. Gene pairs are defined to be functionally co-expressed as long as they have at least one pathway in common.

### Graphical lasso

Graphical lasso [19] was used to infer the gene co-expression network, using the implementation in QUIC R package [20]. We used a set of 32 points from 0.2 to 0.82 as $\rho$ parameter in graphical lasso, and used the default settings for the other parameters. Only the percentage of connections above 0.05% are used in the plot. We subsample the data to 4000 genes to reduce run time of the graphical lasso.

### Remove batch effect and generate gene correlation matrix

We applied principal component analysis to address the confounding noise from batch effects to the expression matrix through removing leading PCs [11]. For each tissue, we scaled the expression matrix such that the expression of each gene has mean 0 and variance 1 across the samples. We regressed out the top PCs and created a matrix of the regression residuals using the WGCNA R package [3].

For each tissue, we considered two approaches to remove PCs from the correlation matrix. In the first approach, we regressed out the top 4 PCs. In the second approach, we regressed out the number of top PCs determined from using the `num.sv` function in the `sva` package [21]. Based on the residuals from these matrices, we generated the gene-gene correlation matrix by calculating the Pearson correlation coefficient of the residuals between genes, for each type of residuals and every tissue.

### A model for correlation in RNA-seq data

Let $Y_1$, $Y_2$ be the data from two observed genes and $Z_1$, $Z_2$ be the unobserved "true" expression variables. We assume that $E(Y|Z) = Z$. If we furthermore assume that $Y_1$ and $Y_2$ are conditional independent given $(Z_1, Z_2)$ (independence of the sequencing noise) we get

$$
\begin{aligned}
\text{Cov}(Y_1, Y_2) &= \text{Cov}(E(Y_1 \mid Z_1, Z_2), E(Y_2 \mid Z_1, Z_2)) + \\
&\quad E(\text{Cov}(Y_1, Y_2 \mid Z_1, Z_2)) \\
&= \text{Cov}(Z_1, Z_2)
\end{aligned}
$$

because of our two assumptions. By multiplying and dividing with the same factors we now immediately get

$$\mathrm{Cor}(Y_1, Y_2) = \mathrm{Cor}(Z_1, Z_2) \frac{\mathrm{sd}(Z_1)}{\mathrm{sd}(Y_1)} \frac{\mathrm{sd}(Z_2)}{\mathrm{sd}(Y_2)}$$

Looking at the standard deviations of the expression level, we get

$$\mathrm{sd}(Y_1) = \sqrt{\mathrm{Var}(Z_1) + \mathrm{E}[\mathrm{Var}(Y_1 \mid Z_1)]} < 1$$

If we make the additional assumption of $\mathrm{Var}(Y_1|Z_1) = Z_1$, we obtain

$$\mathrm{sd}(Y_1) = \sqrt{\mathrm{Var}(Z_1) + \mathrm{E}(Z_1)}$$

Forming the ratio and dividing both the numerator and the denominator by $\sqrt{\mathrm{E}^2(Z_1)}$ yields

$$\frac{\mathrm{sd}(Z_1)}{\mathrm{sd}(Y_1)} = \sqrt{\frac{\mathrm{Var}(Z_1)}{\mathrm{Var}(Z_1) + \mathrm{E}(Z_1)}} = \sqrt{\frac{\mathrm{CV}^2(Z_1)}{1/\mathrm{E}(Z_1) + \mathrm{CV}^2(Z_1)}}$$

These assumptions hold when $Y_1|Z_1 \sim \mathrm{Poisson}(Z_1)$ and can easily be modified to include non-random library sizes. However, we apply these results to $\log_2(\mathrm{RPKM})$ (including after removal of PCs), and it is worthwhile to consider these assumptions in general. The first assumption, that $E(Y|Z) = Z$ is necessary to recover the correlation, but can easily be relaxed to the conditional expectation being an affine function of $Z$; $E(Y|Z) = \alpha + \beta Z$ where $\alpha$ will disappear and $\beta$ be part of the adjustment factor. The second assumption, $\mathrm{Var}(Y_1|Z_1) = Z_1$, is unnecessary—the main requirement is that $E[\mathrm{Var}(Y_1|Z_1)]/E^2(Z_1)$ vanishes when $E(Z_1)$ is large.

## Variance stabilizing transformation

`varianceStabilizingTransformation` function in `DESeq2` package was used on the gene expression matrix of tissue adipose subcutaneous after removing 4 principle components.

## Using combat

`ComBat` function in `sva` package was used in the gene expression matrix on the $\log_2(\mathrm{RPKM})$ scale of tissue adipose subcutaneous, using the date of nucleic acid isolation batch (SMNABTCHD) as the covariate.

## Spatial quantile normalization

To correct the mean-correlation bias in the gene correlation matrix, we developed a method called spatial quantile normalization (SpQN) that removes the difference in "local" distribution of correlations across the gene correlation matrix.

Binning the correlation matrix into disjoint grids and normalizing them separately could result in artifacts. The within-bin variance of distribution could result in imprecise estimation of the local distribution, and therefore the normalization would lead to high within-bin variance. On the other hand, using small bins could result in insufficient sample size for approximating local distribution. We therefore introduce a quantile normalization method that could address this binning problem.

For each tissue, we used the expression matrix (on the $\log_2(\mathrm{RPKM})$ scale), with the genes sorted according to the expression level (average expression across samples). We grouped the

genes into 10 separate disjoint bins and numbered them by 1,2,…,10, with the expression level from low to high. For the corresponding correlation matrix with the genes sorted in the same way, we stratified the matrix into 10 by 10 disjoint equal-size grids, and numbered the grids as $(i, j)$ using the expression level for the $i^{th}$ and $j^{th}$ bin. The selection of target distribution, $F_{ref}$, can be arbitrary, but we used the empirical distribution of grid (9,9).

The set of disjoint submatrices $X_{i,j}$ and the larger submatrices $Y_{i,j}$ that embeds $X_{i,j}$ were assigned based on the preset parameters—number of bins (written as $n_{group}$) and size of larger bins (written as $w$), with default settings $n_{group} = 60$ and $w = 400$.

$\{Y_{i,j}\}$ is assigned to be equal-size, equal-distance and overlapped bins that covered the correlation matrix, with bin size $w$ and distance $d = (n_{gene} - w)/(n_{group} - 1)$, where $n_{gene}$ is the number of genes. $\{Y_{i,j}\}$ can be written as

$$Y_{i,j} = \{\{g, g'\} : \quad (i - 1)d + 1 < g \le (i - 1)d + w,$$
$$(j - 1)d + 1 < g' \le (j - 1)d + w\},$$

for $i, j = 1, 2, \ldots, n_{group}$.

The set of submatrices $\{X_{i,j}\}$ is assigned to be disjoint and same-distance bins, with distance equals to that of $Y_{i,j}$, written as

$$X_{i,j} = \quad \{\{g, g'\} : n_1(i) < g \le n_2(i),$$
$$n_1(j) < g' \le n_2(j)\},$$
$$i, j = 1, 2, ..., n_{group},$$

where

$$n_1(x) = \begin{cases} 0, & \text{if } x = 1 \\ n_2(x - 1), & \text{otherwise} \end{cases}$$

$$n_2(x) = \begin{cases} d/2 + w/2, & \text{if } x = 1 \\ n_{gene}, & \text{if } x = n_{group} \\ n_1(x) + d, & \text{otherwise.} \end{cases}$$

Using using the empirical distribution of $Y_{i,j}$, we estimated local distribution for $X_{i,j}$,

$$F_{emp}(X_{i,j}) = F_{emp}(Y_{i,j}), \quad i, j = 1, 2, 3, ..., n_{group},$$

We mapped the correlations within each disjoint bin to the corresponding quantiles in the target distribution,

$$\tilde{X}_{i,j} = q_{target}(F_{emp}(X_{i,j})) \ ,$$

where $q_{target}$ is the quantile in the target distribution, and $\tilde{X}_{i,j}$ is the correlations in $X_{i,j}$ after quantile normalization. If $Y_{i,j}$ is the same as $X_{i,j}$ this is the same as quantile normalization. As $Y_{i,j}$ gets larger than $X_{i,j}$, $F_{emp}(X_{i,j})$ is only approximately uniform and the corrected correlations $\tilde{X}_{i,j}$ are only approximately following the target distribution.

## Results

### The distribution of gene-gene correlations depends on gene expression level

Crow et al. [13] observed that highly expressed genes tend to be more co-expressed, an observation we later re-discovered [9]. Related, differential expression has been shown to confound

differential co-expression [14]. Here, we use bulk RNA-seq data from 9 tissues from the GTEx project [15] to further explore the relationship between gene expression and gene-gene co-expression.

Gene counts were converted into $\log_2$-RPKMs (Materials and methods). Starting with 19,836 protein-coding genes and 7,036 long non-coding RNAs (lncRNAs) in each tissue, we kept genes with a median expression above zero (on the log-RPKM scale, Materials and methods), leaving us with 10,735–12,889 expressed genes per tissue (of which 95%-98% were protein-coding). Removing a set of top principal components (PCs) from the correlation matrix has been shown to remove unwanted variation in co-expression analysis [11]. We computed the gene-gene correlation matrix of the $\log_2$-RPKM values and removed unwanted variation by removing the top 4 PCs. The number 4 was chosen based on our previous analysis [9], where we used positive and negative control genes to determine this number. An alternative approach suggested by Parsana et al. [11] is to remove the number of PCs according to the estimated number of surrogate variables using SVA [21–23] with the number of PCs ranging from 10 to 30 in these same tissues—the impact of choosing a different number of PCs will be examined below.

For each tissue, we sorted genes according to their expression level (average $\log_2$-RPKM across replicates of that tissue) and grouped them into 10 bins of equal size. We number these gene expression bins from 1 to 10, with 1 being lowest expression. Then, we divided the gene-gene correlation matrix into a 10x10 grid of 100 non-overlapping submatrices numbered as ($i$, $j$) using the expression level for the $i^{th}$ and $j^{th}$ bin (Fig 1). The use of a 10x10 grid is somewhat arbitrary, but it ensures a substantial number of correlations inside each submatrix.

As an example, we begin by exploring one tissue, specifically adipose subcutaneous. In this tissue, the distribution of correlations within the 10 diagonal submatrices are all centered around 0, but their variance increases with expression level (Fig 2a). A robust estimate of the spread of a distribution is given by the interquartile-range (IQR), the difference between the 25% and 75%-quantiles. We can depict the IQR across the binned correlation matrix, forming what we term a 2D boxplot (Fig 2b). This reveals the *mean-correlation* relationship: namely, the IQR of each submatrix is associated with the average expression level in the two corresponding bins, and more specifically, the IQR is approximately dependent on the minimum of the two expression levels (Fig 2c).

In Freytag et al. [10], it is suggested that genes selected at random should be uncorrelated, and the authors verify this to be true empirically in multiple datasets. Another way of stating this assumption is that the true gene-gene correlations between random genes should be close to zero, suggesting that the true correlation matrix is sparse. Therefore, under this model we expect that the observed distribution of gene-gene correlations to be made up of mostly true correlations of zero coupled with some "background" (or random) noise centered around 0. This is assumption is supported by the observed distributions of pairwise correlations depicted in Fig 2a, which are largely symmetric around 0—one or two expression bins have a slight location shift away from 0. However, in Fig 2a we also see that the background distribution depends on gene expression level.

In addition to the behaviour of the true correlations, there is also the impact of measurement uncertainty. In general, higher expressed genes ought to have less noise when estimating their expression level and also their associated pairwise correlation, at least compared to lowly expressed genes. This suggests that measurement noise ought to be decreasing as expression level increases, the opposite trend of what we observe.

The mean-correlation relationship is still present in data processed with a variance stabilizing transformation, a transformation which aims at removing the known mean-variance relationship in RNA-seq data (Fig A in S1 Text).

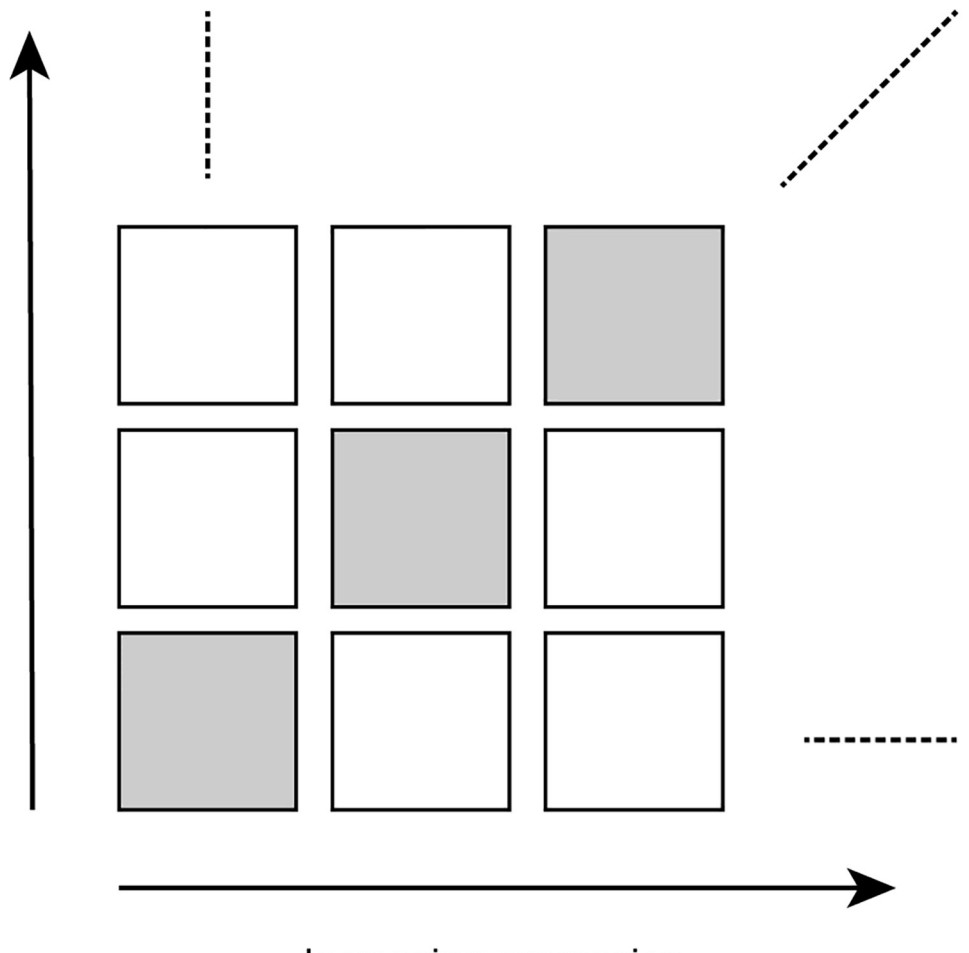

**Fig 1. Partitioning the gene-gene correlation matrix.** Genes are sorted and binned according to increasing expression level, and the correlation matrix is partitioned into 10x10 non-overlapping submatrices of equal size. The diagonal submatrices are indicated by gray shading.

### The mean-correlation relationship biases co-expression analysis

In co-expression analysis, the goal is often to identify biologically meaningful gene pairs with what is assumed to be high correlations (signal) compared to random gene pairings, usually with low correlation (noise). Therefore, a common first step is to separate these highly correlated gene pairs, or clusters (sometimes called modules), from the lowly correlated gene pairs. There are multiple approaches for this, including thresholding the correlation matrix, using weighted gene correlation analysis (WGCNA) [3] or using the graphical LASSO, which operates on the precision matrix; the inverse of the covariance matrix [4].

To visualize the (possible) signal component of the correlation matrix, we overlay the full (background) distribution with the distribution of the top 0.1% of the correlations (signal) within each expression bin. In addition to the mean-correlation relationship in the background distribution, we see that the signal component of the correlation distributions is also dependent on the expression level (Fig 3a). We observe that—conditional on the expression level—the signal and background distributions are well separated, but the point of separation

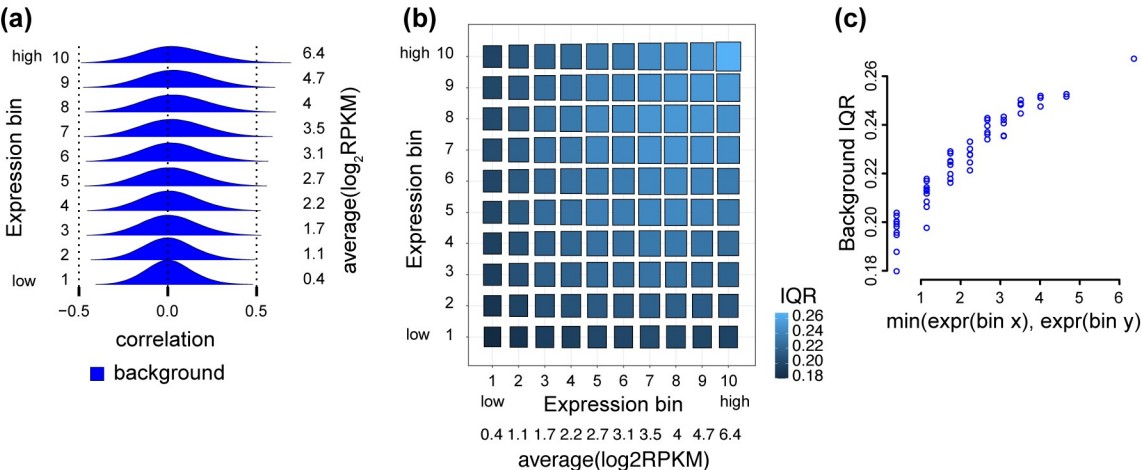

**Fig 2. The mean-correlation relationship between gene expression level and the distribution of observed gene-gene correlations.** The distribution of Pearson correlations of gene pairs using 350 RNA-seq samples from adipose subcutaneous tissue, with 4 PCs removed. **(a)** Densities of the Pearson correlation between gene pairs stratified by overall expression (10 bins ranging from low to high expression). Average expression level for each expression bin is given by the values to the right of the densities. **(b)** A 2D boxplot where each box represents the IQR of the Pearson correlations between all genes (termed background IQR) in a submatrix of the correlation matrix corresponding to two bins of expression. **(c)** The relationship between IQRs of the Pearson correlations between all genes in a submatrix (y-axis), and the minimum between the average expression level of the two bins associated with the submatrix (x-axis).

between the two distributions depends on the expression level. As a consequence, thresholding the correlation matrix for only the top 0.1% of correlations to identify the possible signal used in network topology will result in an over-representation of highly expressed genes (Fig 3b), which elucidates the previously unexplained observation from [13] and [9] that highly expressed genes tend to be more highly co-expressed.

Furthermore, we do not observe this bias towards high expression when examining the expression level of gene pairs which are involved in known protein-protein interactions (PPIs) (Fig 3c) or known regulatory pathways (Fig 3d). This strongly suggests that the observed bias

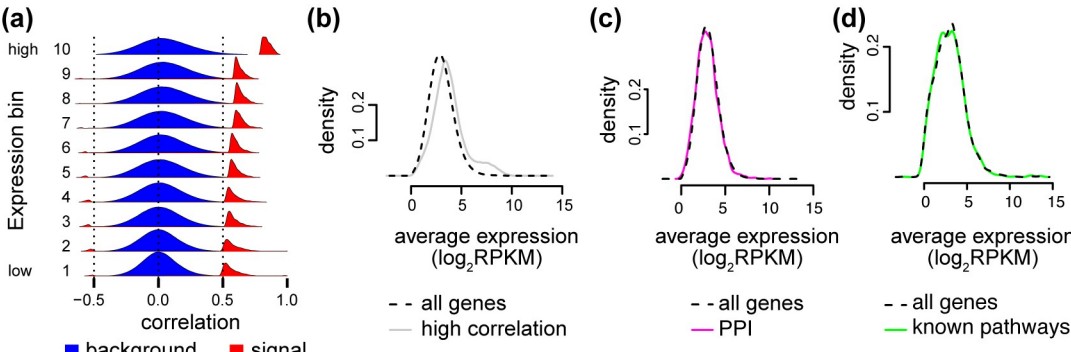

**Fig 3. The mean-correlation relationship leads to expression bias.** Same data as Fig 2. **(a)** Like Fig 2a, but supplemented with the densities (scaled differently from the background densities) of the top 0.1% of the correlations in each expression bin, representing possible signal. **(b)** We calculate the average expression level between two genes involved in a gene-gene correlation, as a measure of the expression level of the pair. The expression level of pairs of genes for either all expressed genes (black) or all gene pairs in the top 0.1% of the correlations (gray). **(c)** The expression level of pairs of genes for either all genes (black) or all gene pairs with a known protein-protein interaction (PPI) (pink).**(d)** The expression level of pairs of genes for either all genes (black) or all gene pairs within a regulatory pathway (green).

is not biological and is unwanted. Here we use the word "bias" to describe that highly correlated genes tend to be highly expressed. We are not using it to describe a potential bias of the empirical correlation estimator.

## A model-based investigation of the mean-correlation relationship in RNA-seq data

To understand the source of the mean-correlation bias, we investigate the consequences of the standard statistical model for bulk RNA-seq data. The negative binomial distribution is widely used in differential expression analysis of bulk RNA-seq data, including in methods such as edgeR and DESeq2 [24, 25]. The negative binomial distribution can be obtained as a Gamma-Poisson mixture as follows. We observe counts $Y$ which reflect the true, but unknown, expression level $Z > 0$ and where $Y \mid Z \sim \text{Poisson}(Z)$. Here, $Y$ represents the expression level filtered by Poisson noise which we believe arise from the counting aspect of sequencing. If the unknown expression level $Z$ follows a Gamma distribution, $Y$ will then follow a negative binomial distribution, but here we will consider a more general model where $Z$ can have any distribution concentrated on the positive real values. It has been experimentally verified that $Y \mid Z = z \sim \text{Poisson}(z)$ for bulk RNA-seq data [26, 27]. In differential gene expression analysis, this model is used separately for each gene, and observations from different RNA-seq samples are considered to be independent.

It is instructive to reflect on how this model can be extended to correlations between genes. Let $Y_1$, $Y_2$ be the data from two observed genes and $Z_1$, $Z_2$ be the unobserved "true" expression variables. We assume that $E(Y \mid Z) = Z$. Under mild assumptions, we can show (Materials and methods)

$$\text{Cor}(Y_1, Y_2) = \text{Cor}(Z_1, Z_2) \frac{\text{sd}(Z_1)}{\text{sd}(Y_1)} \frac{\text{sd}(Z_2)}{\text{sd}(Y_2)}$$

This implies that the observed correlation is equal to a scaled version of the "true" correlation, with a gene-dependent adjustment factor. The adjustment factor is essentially driven by how much extra variation $Y$ introduces on top of $Z$. Considering the adjustment factor, we can show (Materials and methods)

$$\frac{\text{sd}(Z_1)}{\text{sd}(Y_1)} = \sqrt{\frac{\text{CV}^2(Z_1)}{1/\text{E}(Z_1) + \text{CV}^2(Z_1)}} < 1$$

with $\text{CV}^2(Z_1)$ being the squared coefficient of variation. This shows that the observed correlations are strictly smaller than the true correlations, with an adjustment factor close to 1 when the expression level of both genes is high.

This model explains why—for genes with a true non-zero correlation—the width of the background distributions decrease with decreasing expression level (the adjustment factors decreases) and suggests that the "true" width of the background distribution is observable for highly expressed genes. Furthermore, it suggests that the background distributions in different submatrices are roughly related through a scaling transformation. However, this argument falls apart if we believe the true expression network to be sparse, ie. that most genes are truely uncorrelated.

To explore whether the background distributions from different submatrices are related by scale transformations, we use quantile-quantile plots (Q-Q plots). If two distributions are related by a scale transformation, the Q-Q plot will be a straight line, with the slope of the line giving the scale parameter. Fig B in S1 Text suggests that a large subset of the submatrices

exhibit scaling differences, but that this is not true across all submatrices. The submatrices with non-linear differences are all at the "boundary" of the correlation matrix, with either very lowly expressed or very highly expressed genes (with a greater proportion of lowly expressed genes exhibiting this behavior).

In summary, this model is at best a partial explanation of the observed phenomena.

## Spatial quantile normalization

To correct for the mean-correlation bias, we developed a spatial quantile normalization method, referred to as SpQN. Here "spatial" refers to the spatial ordering of expression along the two dimensions of the correlation matrix. The objective is to achieve the same "local" distribution of correlations across the matrix. In other words, different submatrices of the gene-gene correlation matrix should exhibit similar distributions. However, unlike quantile normalization [28–30], our method does not mathematically guarantee that different submatrices end up with the same empirical distribution, although our experiments suggest that this is approximately true. SpQN takes as input a correlation matrix and a gene-specific covariate (here: expression level) and outputs a normalized correlation matrix. The gene expression matrix is not modified.

To explain our approach, recall that standard quantile normalization works by transforming observed data $X$ using

$$Y = q_{\text{target}}(F_{\text{emp}}(X))$$

where $F_{\text{emp}}$ is the empirical distribution function for $X$ and $q_{\text{target}}$ is the quantile function for a suitably chosen target distribution. In its original formulation of quantile normalization, $q_{\text{target}}$ was chosen empirically as the average quantile distribution across samples.

Consider a submatrix $X_{i,j}$ of the gene-gene correlation matrix (from the $i^{th}$ and $j^{th}$ ordered expression bins). Instead of using the empirical distribution ($F_{\text{emp}}(X_{i,j})$) of $X_{i,j}$ to form a distribution function—as is done in standard quantile normalization—we use a larger submatrix $Y_{i,j}$ enclosing $X_{i,j}$ as the basis of the empirical distribution function (Fig 4a). This implies that when two submatrices, $X_{i,j}$ and $X_{i+1,j}$, are adjacent, their enclosures $Y_{i,j}$ and $Y_{i+1,j}$ are overlapping, which ensures a form of continuity in their associated distribution functions (Fig 4b). Because $X_{i,j}$ and $X_{i+1,j}$ are non-overlapping, each point in the correlation matrix is associated with a unique empirical distribution function. We employ this approach to avoid discontinuities in the normalization functions at the boundary of the submatrices. The size of $X_{i,j}$ and $Y_{i,j}$

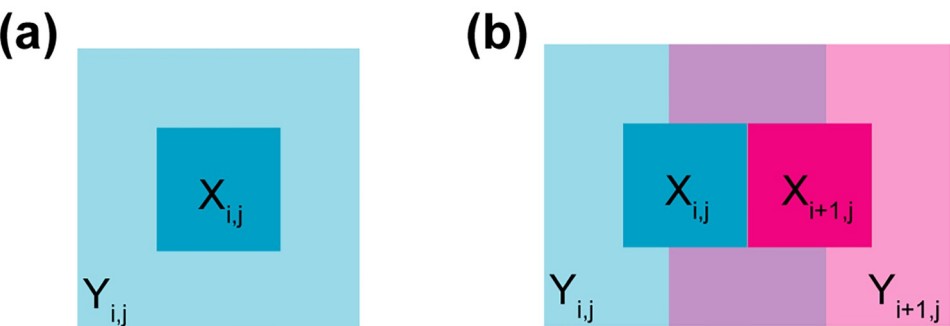

**Fig 4. Spatial quantile normalization explained.** (a) A submatrix $X_{i,j}$ of the correlation matrix, and its enclosing submatrix $Y_{i,j}$. (b) Two directly adjacent, non-overlapping, submatrices $X_{i,j}$, $X_{i+1,j}$ and their enclosing, overlapping, submatrices $Y_{i,j}$, $Y_{i+1,j}$. The enclosing submatrices $Y_{i,j}$, $Y_{i+1,j}$ are used to form the empirical distribution functions $F_{\text{emp}(Y_{i,j})}$, which are then applied to the non-overlapping submatrices as $F_{\text{emp}(Y_{i,j})}(X_{i,j})$.

is chosen by the user and controls the degree of smoothing, not unlike a bandwidth parameter for a density estimator. In our application, we use $60 \times 60$ outer enclosures, with each outer enclosure containing approximately $400 \times 400$ gene-gene correlations and each inner enclosure containing approximately $200 \times 200$ gene-gene correlations. We found this setting to work well across our applications.

The choice of target distribution (with quantile function $q_{target}$) can be arbitrary, but in this application we require that its support of the distribution should be contained in $[-1, 1]$. We recommend a specific submatrix of the correlation matrix to be the target distribution, specifically the (9, 9)-correlation submatrix (out of a 10x10 binning). This is based on the insights from the preceding section, which suggests that the observed pairwise correlation between two genes is equal to the unobserved pairwise correlation of their expression levels, provided the two genes are highly expressed and thereby less affected by technical noise. We avoid the top right submatrix (10, 10), because it may contain a wide range of expression levels as we form submatrices of equal size; note the unusual behavior of the top right submatrix in Fig B in S1 Text. For more details on SpQN, we refer the reader to the Materials and Methods section.

Note that mean expression is only used to sort the bins from the correlation matrix. We use expression level as a measure of distance between genes so that genes with similar expression level are close (see Discussion for additional comments on the generality of SpQN).

## Spatial quantile normalization removes the mean-correlation relationship

Applying spatial quantile normalization to the 350 RNA-seq samples from adipose subcutaneous tissue in GTEx, we found that it corrects both the background and signal distributions (Fig 5a) across the correlation matrix, thereby removing the mean-correlation relationship (Fig 5b and 5c). In addition, our normalization method removes the bias towards highly expressed genes when using thresholding to identify highly co-expressed genes (Fig 5d, compared to Fig 3b).

Our observations hold true across a diverse set of GTEx tissues (Fig 6a, Figs C-D in S1 Text). For each of the 10x10 submatrices of the tissue-specific correlation matrix, we compute

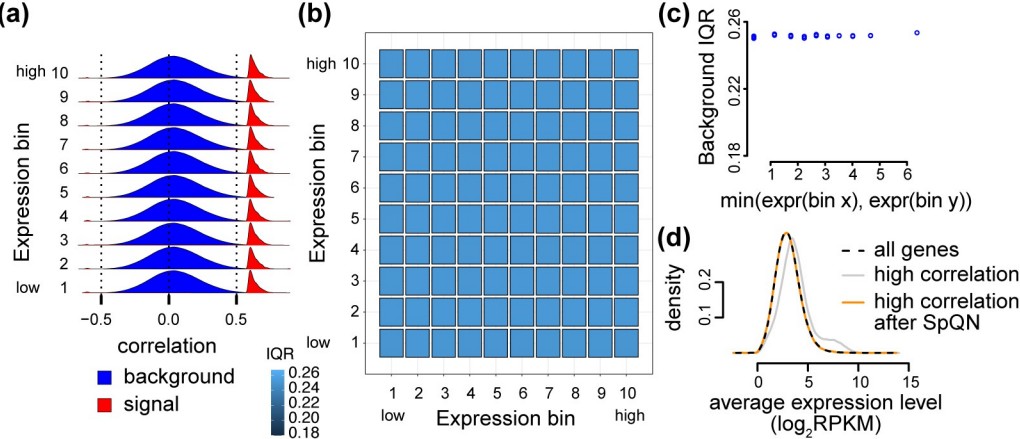

**Fig 5. Spatial quantile normalization removes the mean-correlation relationship.** Same data as Fig 2, but after applying spatial quantile normalization (SpQN). **(a)** Like Fig 3a, i.e. densities of the Pearson correlation between all genes within each of 10 expression bins (background) as well as the top 0.1% correlations (possible signal). **(b)** Like Fig 2b, i.e. IQRs of Pearson correlations between genes in each of 10 different expression levels. **(c)** Like Fig 2c, i.e. the relationship between IQR of gene-gene correlation distribution and the lowest of the two expression bins associated with the submatrix. **(d)** Like Fig 3b, i.e. the expression level of pairs of genes in different subsets (all genes (black), genes above the 0.1% threshold with (orange) and without SpQN (gray)).

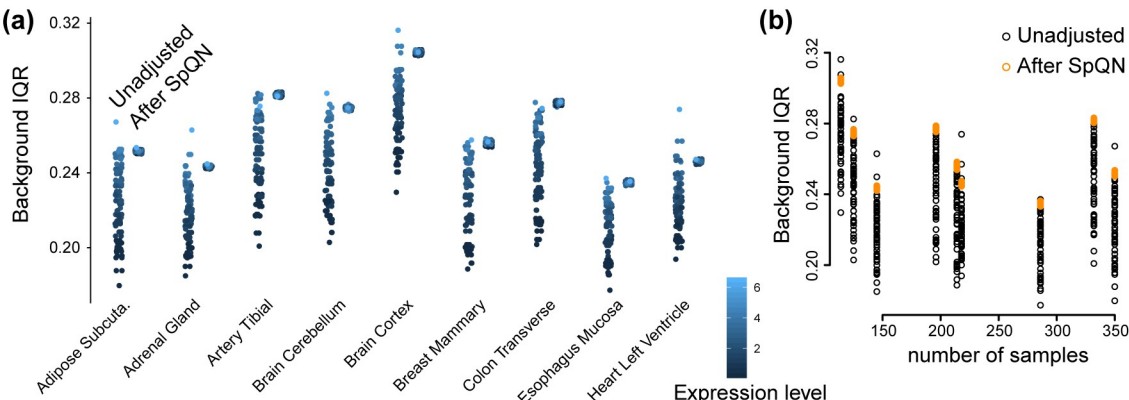

**Fig 6. IQR of gene-gene correlation distributions in each bin for 9 tissues.** RNA-seq data from [15] from 9 tissues with 4 PCs removed. A point in this figure corresponds to one submatrix in a given gene-gene correlation matrix for each tissue, before and after SpQN. **(a)** Background IQR for unadjusted (left smear) and SpQN-adjusted (right smear) gene-gene correlation distributions for all expression bins across 9 GTEx tissues. Color indicates expression level. **(b)** The relationship between sample size for a tissue and background IQR for correlation distributions before and after SpQN adjustment.

an IQR and we consider the distribution of the IQRs before and after applying spatial quantile normalization (Figs C-D in S1 Text). We observe that using our approach makes the IQRs similar across the correlation submatrices, as the width of the boxplots is smaller for SpQN compared to pre-SpQN (Fig 6a) across all 9 tissues. We note that—because of our choice of target distribution—the IQRs of most of the background distributions increase following spatial quantile normalization. We furthermore note that the pre- and post-SpQN range of IQR is tissue dependent, but this is also driven by differences in sample sizes for the different tissues (Fig 6b). Below, we show that the relationship between sample size and IQR range becomes stronger when we remove a tissue-specific number of principal components.

To highlight the impact of our method on biological relationships, we focus on transcription factors, which have been found to be relatively lowly expressed [9, 31]. As transcription factors are an important class of regulatory genes, there is substantial interest in identifying co-expression between transcription factors and other genes.

To quantify the impact of our method on transcription factor co-expression, we use a comprehensive list of 1,254 human transcription factors [32]. For each of our 9 exemplar tissues, we again threshold the correlation matrix and ask how many edges involve transcription factors with and without the use of SpQN. Fig 7a displays the percent increase in edges involving transcription factors following SpQN for various signal thresholds (ranging from 0.1% to 3%) of the correlation matrix for a single GTEx tissue (Additional tissues are depicted in Figs E and F in S1 Text). This result shows an overall increase in edges involving transcription factors. Next, we computed the same percent change, but using protein-protein interactions involving transcription factors. We note that this is a flawed measure as protein-protein interactions are not the same as co-expression and because this analysis at best identifies co-factors and not downstream targets of the transcription factors. We observe an overall increase in edges involving transcription factors, but the increase—as expected—is for lowly expressed genes, whereas highly expressed genes show a decrease. This is partly explained by the zero-sum nature of calling edges based on a fixed percentage of interactions. We conclude that there is some evidence that SpQN improves the inference of interactions involving transcription factors, but this may come at the cost of decreased performance for highly expressed genes and that the overall performance depends on the expression distribution of the genes of interest.

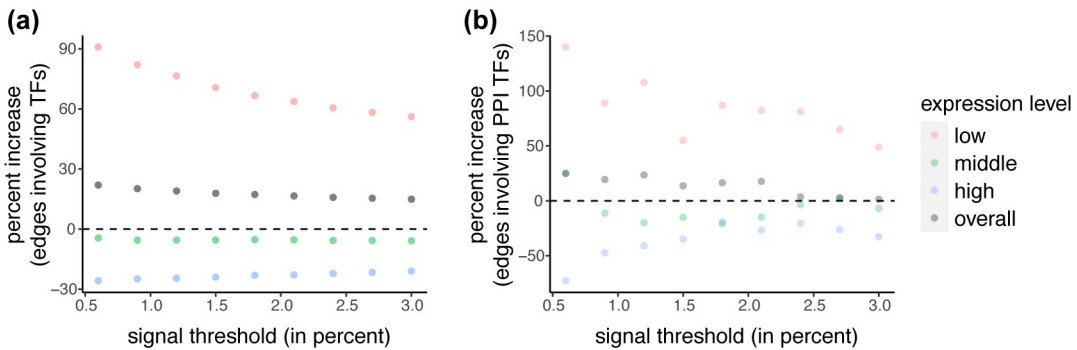

**Fig 7. The impact of SpQN on transcription factor co-expression.** Data is from adipose subcutaneous from GTEx. The percent increase in the number of edges (y-axis) identified after thresholding (x-axis) the correlation matrix. **(a)** Edges involving transcription factors. **(b)** Edges between genes with protein-protein interactions, where one of the involved genes is a transcription factor. Additional tissues are depicted in Figs E and F in S1 Text.

## Expression bias in co-expression signals is influenced by the size of the network

In our analysis so far (with the exception of the transcription factor analysis in the previous section), we have fixed the size of the network by using a fixed signal threshold based on the top 0.1% of correlations. It is natural to ask whether changing this threshold could change our conclusions. Especially, because the limit of a fully connect network (equivalent to a threshold of 100%) will per definition show no difference between the background and the signal distribution.

To quantify the impact of signal threshold on the expression bias of co-expression analysis, we compared the bias across 40 different thresholds ranging from 0 to 3%. For each of the 9 tissues, the bias towards high expression genes exists among all the thresholds except a few extreme small values (Fig 8, Fig G in S1 Text with 4 PCs removed, Fig H in S1 Text for using SVA to estimate the number of PCs to remove). For most tissues, the biases decreases with the increase of the threshold (as expected), although the bias never disappears. For the two tissues showing increasing bias, the increase in bias appears to eventually stop for larger network sizes. For all network sizes, there is no bias after applying SpQN.

## Mean-correlation relationship biases results using the graphical lasso and can be corrected by SpQN

So far, we have constructed networks by thresholding the correlation matrix, a simple and interpretable method. However, SpQN is compatible with any network inference method which takes a correlation matrix as input. To show the versatility of SpQN, here we assess the combination of SpQN with the graphical lasso [19, 33], which is popular in co-expression analysis [11]. Because the graphical lasso works with the *inverse* of the correlation matrix, it is not straightforward that results from the graphical lasso is biased by the mean-correlation relationship. The graphical lasso has a tuning parameter (here denote by $\rho$), which controls the amount of regularization. We calculated the average expression level of genes that are part of the resulting network, using 40 different values of the tuning parameter $\rho$, which controls the size of inferred network (using only 4,000 randomly sampled genes to decrease computational time). We only kept those networks with signals higher than 0.05%. By comparing to the average expression level of all genes, we can assess a the expression bias in the network. Instead of displaying $\rho$, which is hard to interpret, we display the size of the inferred network.

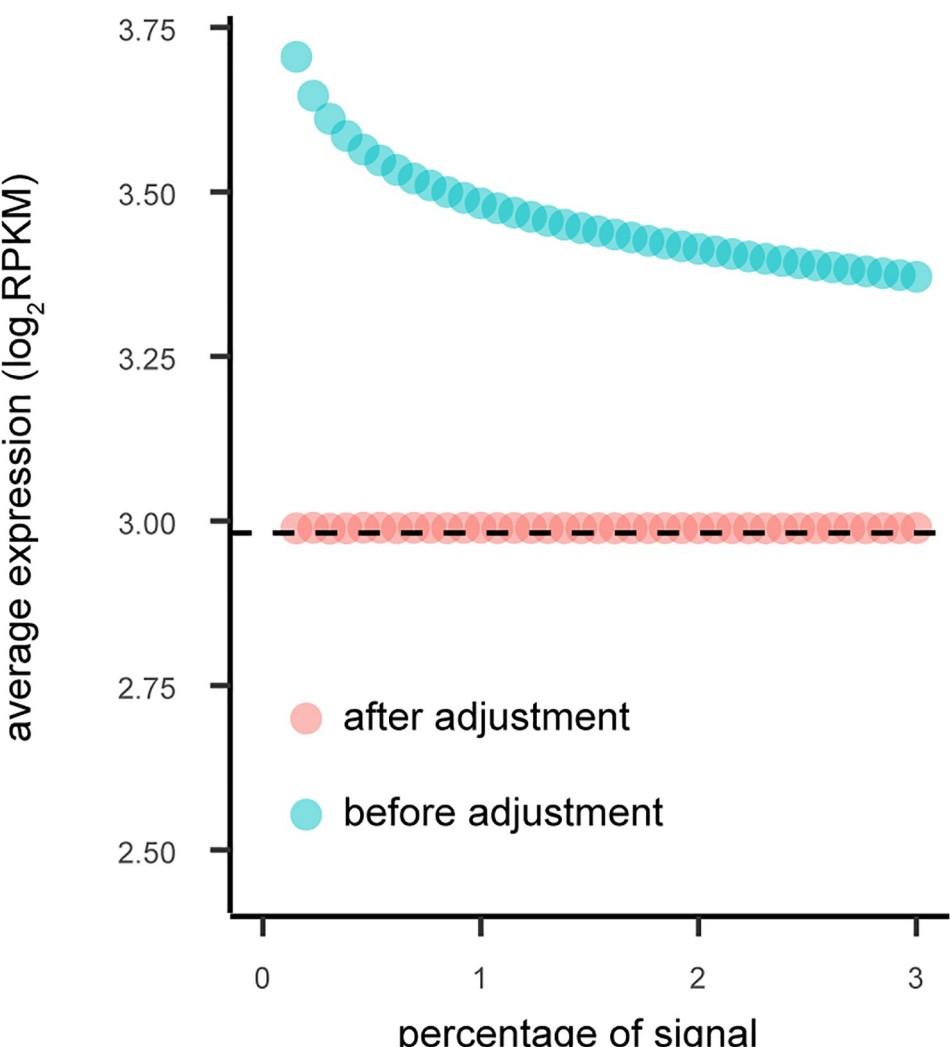

**Fig 8. The relationship between the signal threshold (in percentage) and the expression level (before and after SpQN adjustment).** We define the co-expression signal threshold (x-axis) as the top percentage of absolute correlation values (ranging between 0 and 3%). For a given signal threshold, we calculate the average expression level (y-axis), before (blue) and after SpQN (pink). The average gene expression level is shown by the dotted black line. Data is from the Adipose subcutaneous tissue with 4 PCs removed, see Fig G in S1 Text for all 9 tissues.

We observe that the graphical lasso exhibits expression bias, which (generally) decreases as the size of the network increases (Fig 9, Fig I in S1 Text for all tissues), with some tissues exhibiting low bias for very small networks. This bias is dramatically reduced by applying SpQN prior to network inference, although we still observe a small dependence on network size. This observation is in contrast to the previous section where the SpQN adjusted co-expression network by thresholding exhibited no bias. We hypothesize this difference is at least partly the result of the graphical lasso operating on the inverse of the correlation matrix.

### Mean-correlation bias in single-cell RNA-seq data

Single-cell RNA-sequencing (scRNA-seq) data exhibit the same mean-correlation relationship as in bulk RNA-seq data. We conclude this based on a re-analysis of an unusually deeply

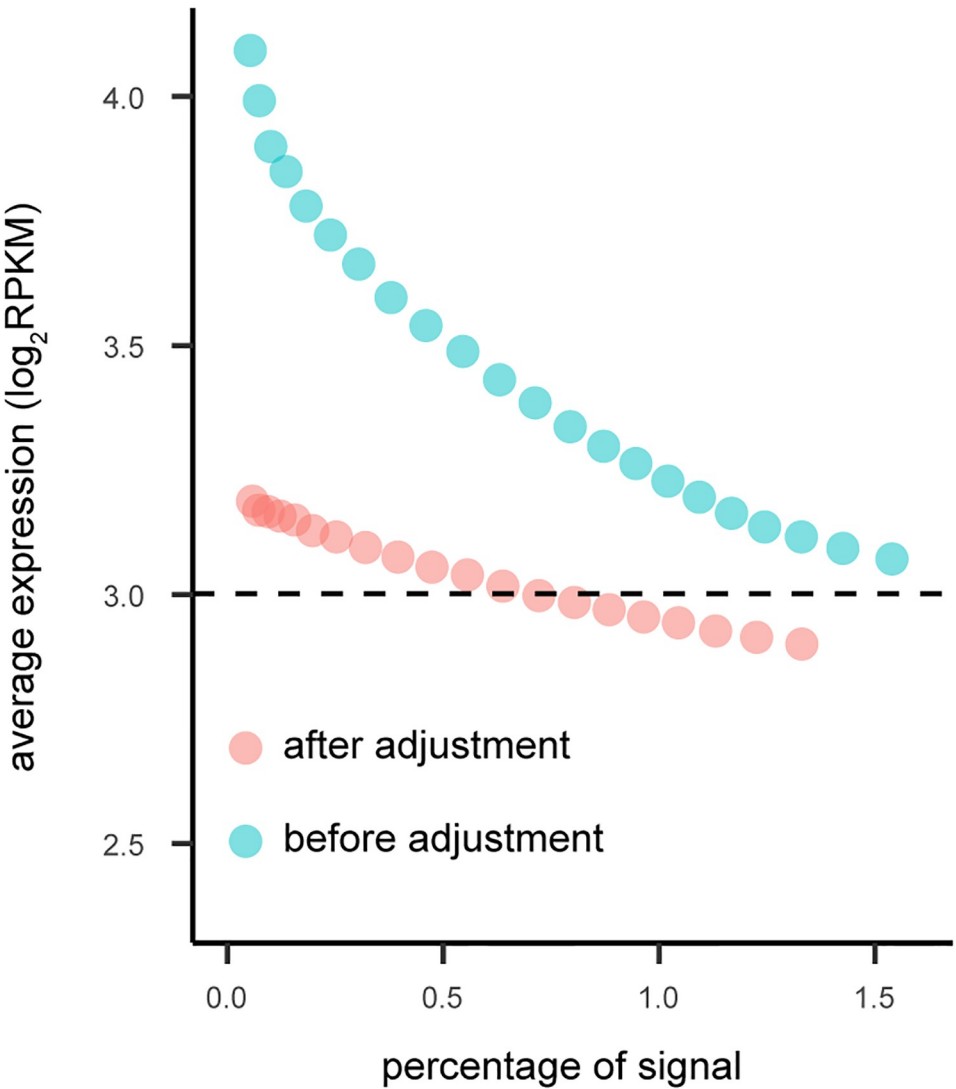

**Fig 9. The expression bias in graphical lasso network inference.** The expression levels of networks inferred by graphical lasso. Different values of the tuning parameter ($\rho$) results in different network sizes (x-axis) with higher values of the tuning parameter leading to smaller networks. The average gene expression is shown by the dotted black line. Data is from Adipose subcutaneous with 4 PCs removed, see Fig I in S1 Text for additional tissues.

sequenced single cell data set [16]. We focus on this particular data set to avoid issues with computing correlation for very sparse data. We kept genes with median $\log_2$(RPKM) greater than 0, leaving 6,915 out of 22,958 genes.

As depicted in Fig 10, this dataset exhibits the same behavior as the GTEx tissues analyzed above (Fig 10a). An over-representation of highly expressed genes following thresholding the correlation matrix at 0.1% is also observed in this scRNA-seq data (Fig 10b). We observe that the bias of correlation towards highly expressed genes is removed following the application of SpQN (Fig 10b). These observations hold true both when the top 4 PCs are removed (Fig 10a and 10b) and when we remove the top 16 PCs (Fig 10c and 10d) found by using SVA to estimate the number of PCs to remove. Unlike the situation for GTEx, we observe that the expression bias is smaller when more PCs are removed (compare Fig 10b to 10d). Together, this

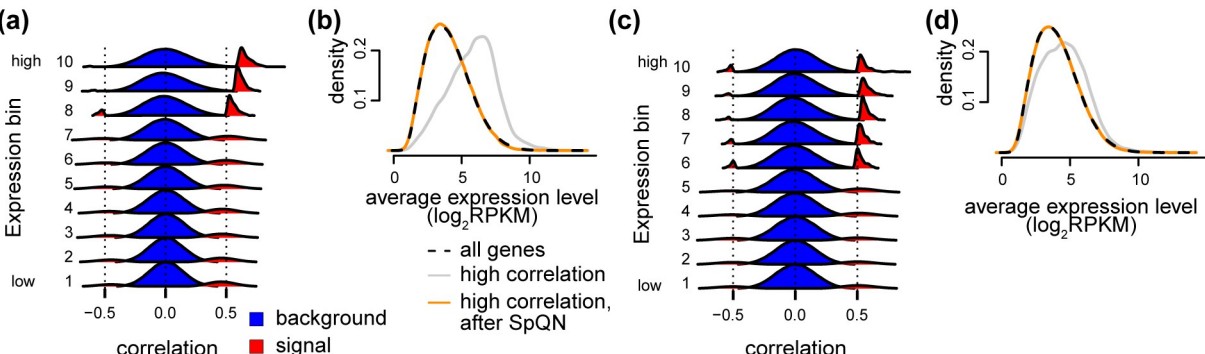

**Fig 10. Mean-correlation relationship in scRNA-seq data.** scRNA-seq data from [16] containing 60 cells, with either 4 PCs removed (a-b) or 16 PCs removed (c-d). The later value is the result of using SVA to estimate the number of PCs as suggested by [11]. **(a)** Like Fig 3a, i.e. densities of the Pearson correlation between all genes within each of 10 expression bins (background, blue) as well as the top 0.1% correlations (possible signal, red). **(b)** Like Fig 5d, i.e. the expression level of pairs of genes in different subsets (all genes (black), genes above the 0.1% threshold with (orange) and without SpQN (gray)). **(c)** Like (a) but for data with 16 PCs removed. **(d)** Like (b) but for data with 16 PCs removed.

suggests that deeply sequenced scRNA-seq data have the same mean-correlation bias as bulk data.

## The impact of removing principal components

Until this point, we have considered data where we have removed batch effects by removing the top PCs. An important analytic component in our assessment of the mean-correlation relationship has been the choice of the background distribution of correlations between all (or random) sets of genes. However, we found that removing PCs can *impact* the background distribution of correlations (first noticed by Freytag et al. [10]). Furthermore, while it is clear that removing PCs removes unwanted variation, it is less clear exactly how many PCs should be removed. Together, this raises the question: to what extent is the mean-correlation relationship dependent on removing PCs prior to calculating the gene-gene correlation matrix? Can we make it go away, simply by removing many PCs? We now explore this question.

First, we focus on properties of the data prior to removing top PCs. Considering the background and (possible) signal distributions for all 9 GTEx tissues, we observe that the background distributions are not necessarily centered around zero (Fig 11a for heart left ventricle, Fig K in S1 Text for all tissues). Based on our arguments that the background distributions ought to be zero-centered under the assumption of a sparse true correlation network, we term this location shift a "background bias". The bias is likely batch (but here we depict this as tissues) dependent and so is its relationship with expression level—contrast "Brain Cerebellum" (high bias with high expression) with "Esophagus Mucosa" (high bias with intermediate expression, low bias with both high and low expression). In addition we observe, as previously described, that both the spread of the background distributions as well as the position of the signal distributions are strongly dependent on expression level (and tissue or batch) (Fig K in S1 Text). Comparing to the same distributions after removing 4 PCs (Fig C in S1 Text), we conclude that removing even a few PCs has a large beneficial effect on the behaviour of the background distributions, including a substantial reduction in background bias.

Next, we focus on removing an increasing number of PCs. If we use the number of PCs recommended by Parsana et al. [11] (Table A in S1 Text), we see similar results (Fig J(a) in S1 Text) compared to removing 4 PCs (Fig 6a). However, both the average and the spread of the IQRs of background distributions prior to applying spatial quantile normalization, are smaller

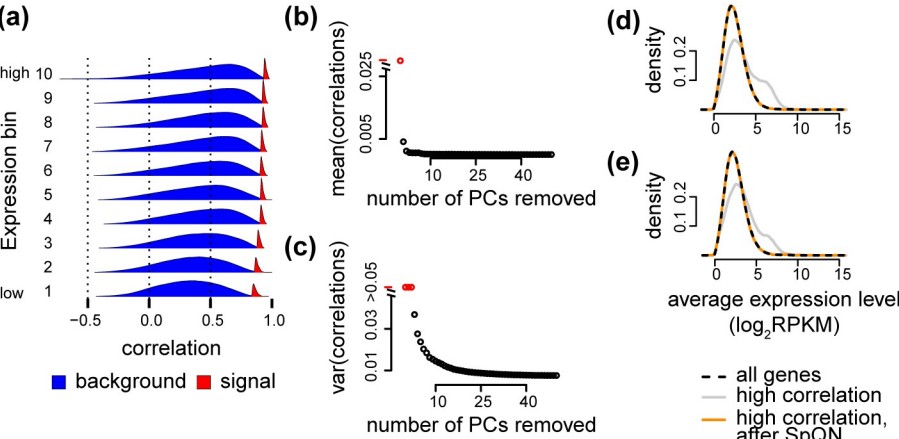

**Fig 11. The impact of removing principal components.** Data from "heart left venticle". **(a)** Background and signal distribution without removing principal components. **(b)** Average bias (median of the 10 background distributions) as a function of PCs removed. **(c)** Average variance (average variance of the 10 background distributions) as a function of PCs removed. **(d)** Average expression after removing 4 PCs. **(e)** Average expression after removing a number of PCs estimated using SVA.

than using 4 PCs. Interestingly, the variation across tissues in IQRs are now more driven by sample size (Fig J(b) in S1 Text) compared to using 4 PCs (Fig 6b).

Finally, we quantified the effect of removing PCs on the bias and spread of the background distributions (Fig 11b and 11c for heart left ventricle, Fig L in S1 Text for all tissues). Bias is reduced across all tissues, although the largest decrease happens with the first few PCs (perhaps up to 10 PCs). The spread of the background distributions are also reduced, although for some tissues there is an inflection point after which the spread increases with higher number of PCs. Together, these observations suggests that removing a high number of PCs may have a substantial positive impact on the mean-correlation relationship.

However, more important is the impact on downstream analysis, particularly the identification of highly co-expressed genes in co-expression analysis. Using our previously described approach of thresholding the correlation matrix, we observe that removing a larger number of PCs has a small impact on the expression bias for highly co-expressed genes (Fig 11d,e for heart left ventricle, Fig M in S1 Text for all tissues), suggesting that focusing on background spread by itself is irrelevant, but that it is more important to evaluate impact on the background distributions relative to the impact on the (possible) signal distributions. Importantly, we observe that spatial quantile normalization removes this expression bias, irrespective of how many PCs were removed.

Many methods have been proposed to remove batch effects in differential expression analysis. To investigate the impact of alternatives to removing principal components, we use Com-Bat [34] to remove the effect of date of nucleic acid isolation batch in the expression matrix prior to constructing the correlation matrix. The correlation matrix exhibits the expected mean-correlation matrix (Fig N in S1 Text). Note the background distributions are not centered, which—based on the evaluations here—suggests that the is a remaining batch effect signal.

## Co-expression in a differential setting

So far, we have examined correlation matrices obtained from considering biological replicates within a condition; we consider this the classic co-expression setting. An alternative is to

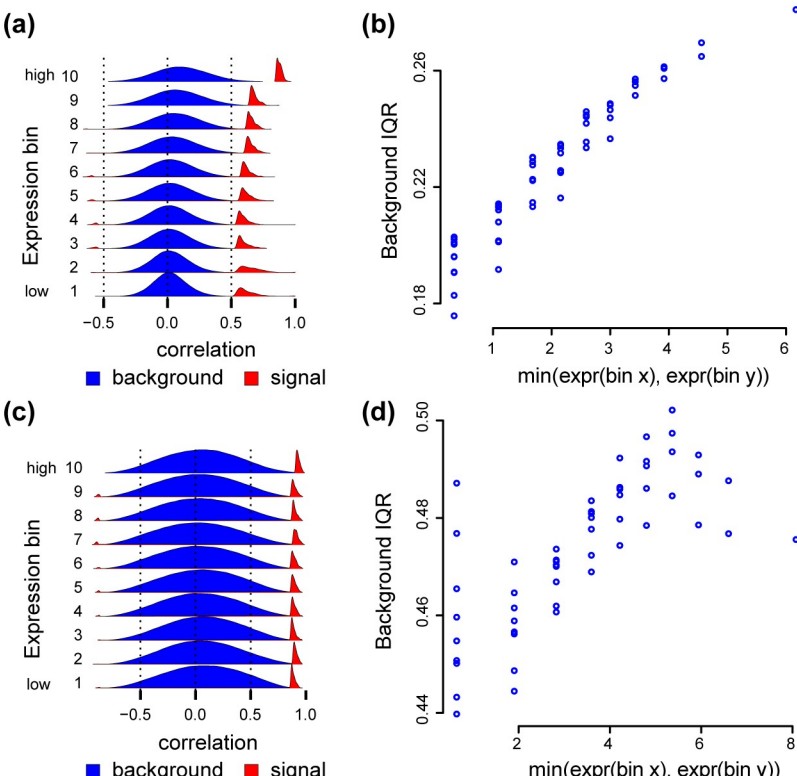

**Fig 12. Mean-correlation in a differential setting.** Data in (a,b): 100 samples were randomly selected from each of 3 GTEx tissues (adipose subcutaneous, adrenal gland and artery tibial) for a total of 300 samples. We removed 4 principal components from the resulting correlation matrix. Data in (c,d): bulk RNA-seq of a time course experiment on drosophila embryonic development with 30 samples. We removed 5 principal components from the resulting correlation matrix. **(a)** Densities of the Pearson correlation between gene pairs stratified by overall expression, for the GTEx data. **(b)** The relationship between IQRs of the Pearson correlations between all genes in a submatrix (y-axis), and the minimum between the average expression level of the two bins associated with the submatrix (x-axis), for the GTEx data. **(c)** Like (a), but for the drosophila data. **(d)** Like (b), but for the drosophila data.

compute correlation matrices where samples are associated with different conditions (including cell types or tissues). We call this the differential setting and this yields a different interpretation of the resulting correlation matrix. An important question is whether such a correlation matrix exhibits a mean-correlation relationship. The answer is not straightforward, because two genes which are both differentially expressed, will be highly correlated, but each gene may be lowly expressed in one condition and highly expressed in another condition.

To examine this question, we consider two scenarios. First, we create a dataset by randomly sampling 100 individuals from each of 3 tissues for a total of 300 samples, a sample size similar to the GTEx tissues previously considered. This pooled dataset exhibits the same mean-correlation relationship as other datasets we have considered (Fig 12a and 12b). Next, we consider data from a time course experiment in the developing *drosophila* embyro, with a total of 30 samples (substantially smaller than other datasets we have considered). This matrix also exhibits a mean-correlation relationship (Fig 12c and 12d), although we make three observations (1) there is substantial variation between the background IQR of different bins associated with the same expression level and (2) the observed background IQRs are substantially larger than observed elsewhere and therefore (3) the change in background IQRs relative to their variation

is smaller then for other datasets we have considered. We hypothesize these observations are the result of the substantially smaller sample size in this dataset.

These two examples show that co-expression analysis in a differential setting may exhibit a mean-correlation relationship. How often this is true, is an open question, but it is easy to assess as part of any co-expression analysis.

## Discussion

A key challenge in gene network reconstruction methods is to select biologically meaningful pairs of co-expressed genes. A standard approach is to select highly correlated gene pairs (possibly signal) compared to lowly correlated gene pairs (background). Here, we demonstrate the existence of a *mean-correlation relationship*, which can bias co-expression analysis resulting in an over-representation of highly expressed genes amongst connected gene pairs. This is not a bias of the estimated Pearson correlation coefficient, but rather a preferential selection of highly expressed genes. This is particularly problematic for genes that are generally expressed at low levels, such as transcription factors. To address this, we have developed a normalization method for the gene-gene correlation matrix that can standardize "local" distributions of the correlations across the matrix. Using nine GTEx tissues with bulk RNA-seq, as well as one deeply sequenced scRNA-seq dataset, we have illustrated how the mean-correlation relationship can be removed from the correlation matrix using spatial quantile normalization (SpQN). Utilizing our method results in a greater number of connections involving transcription factors, an important class of regulatory genes. However, this increase may come at the expense of down prioritizing connections between highly expressed genes. The total benefits of SpQN on overall biological insight is likely to be impacted by the (unknown) expression distribution of the network of interest. For this reason, we suggest that users do not blindly apply the method.

The mean-correlation relationship was first, to our knowledge, described by Crow et al. [13]. In this work, the authors also consider solutions to this problem. Specifically, they consider the problem of assessing connectivity of a fixed set of genes, and recommended that this fixed set of genes is compared to random genes with the same expression level, which will account for the mean-correlation relationship. However, it is not obvious how to generalize this approach to general network inference. This is the problem SpQN addresses.

A common goal of normalization methods in genomics is to increase the between experiments reproducibility. This is not the goal of SpQN. The bias we observe is associated with gene expression level and the expression ranking of different genes is highly reproducible between experiments [35], implying that the bias itself is reproducible. Instead, the goal of SpQN is to increase the representation of lowly expressed genes in co-expression analyses, which—as a result of the bias we observe—is underrepresented in inferred networks. This point of view is supported by [13] who found that expression bias is present even in networks aggregated across experiments.

The magnitude of the bias also depends on the size of the inferred network, because a fully connected network will (by definition) have no bias. It also depends on which genes are selected to be part of the network. Based on our analyses, we do not observe much bias amongst the highest expressed genes. Consider a network construction method which starts by selecting (say) the 20% highest expressed genes and then perform network inference. We would not expect that the constructed network will show much association between expression level and connectivity, since this only become apparent once enough lowly expressed genes have been included. In our view, focusing network inference on the highest expressed genes, is not solving the problem, but merely by-passing it.

We have shown that SpQN removes the expression bias in gene co-expression analysis for networks constructed both by thresholding the correlation matrix and by graphical lasso. SpQN is compatible with any co-expression method that operates on the correlation matrix, such as WGCNA. However, applying SpQN results in a correlation matrix, which is typically point-wise greater than the input matrix; in other words, the correlations increase. Depending on the choice of network inference method, this might have a dramatic impact on the properties of the resulting network.

In our examples we have focused on 9 tissues from GTEx. We find it noteworthy—and perhaps surprising—that almost all high correlations are positive. This happens both for unprocessed data, data where we have removed 4 (or more) principal components and data processed with SpQN.

We have presented a simple model to provide intuition for the cause of the expression bias; an extension of the standard model for the analysis of individual genes for differential expression in bulk RNA-seq data extended to gene pairs. This model shows that the observed pairwise correlations are a perturbed version of the true pairwise correlations of interest, and this perturbation is caused by the count-based nature of bulk RNA-seq data. Ultimately, we conclude that a non-parametric correction approach is better suited to address properties of the observed data. We include the model-based motivation because it helped sharpen our thinking and highlights the difference between the observed and true correlations of interest.

We have investigated the impact of removing PCs on the mean-correlation relationship, which revealed a number of interesting observations. We made this investigation because we (and others) observed that removing PCs changes the background distribution of the data. First, importantly, while removing top PCs impacts the spread and bias of the background distributions, we observe a similar bias towards highly expressed genes when removing 4 PCs as when removing 10–30 PCs. Second, the different patterns of expression dependence in the background distributions are likely to represent tissue-dependent batch effects. This therefore serves to illustrate how batch effects impact co-expression analysis. An interesting question is what forces create these different patterns. Altogether, our work reinforces the message from Parsana et al. [11] that removing PCs is good practice, although exactly how many components to remove is still an open question.

Finally, we highlight two points regarding the generalizability of the core idea of SpQN. First, the formulation (and software) merely requires the correlation matrix to be sorted according to a confounding variable; the fact that in this application the confounding variable is mean expression is not critical. Second, the core idea of SpQN extends far beyond correlation matrices. Specifically, the way we propose to use neighborhoods of different sizes to define inner and outer enclosures, only requires some sense of distance to define the neighborhoods. This suggests that the idea of SpQN can be applied to any kind of data with a natural sense of distance between observations, such as time series data or spatial data.

Together, this work shows the importance of assessing and addressing mean-correlation bias in co-expression analysis, and provides the first method for doing so.

## Supporting information

**S1 Text. Fig A: Variance stabilization does not remove the mean-correlation relationship**. Same raw data as Figs 2 and 3, but we apply a variance stabilizing transformation (as implemented by DESeq2) followed by removing 4 principal components. **(a)** Like Fig 3a, i.e. densities of the Pearson correlation between all genes within each of 10 expression bins (background) as well as the top 0.1%. **(b)** Like Fig 2c, i.e. the relationship between IQR of gene-gene correlation distribution and the lowest of the two expression bins associated with

the submatrix. **Fig B: Distribution comparison for different submatrices of the observed correlation matrix (after removing the top 4 PCs)**. Same data as Fig 2. Quantile-quantile plots comparing the distribution of Pearson correlations in various $(i, i)$ submatrices (y-axis) to the $(9, 9)$ submatrix(x-axis). **Fig C: The background and signal components depends on expression level across many tissues**. Data is 9 different GTEx tissues, all with 4 PCs removed. Distributions of Pearson correlations for genes within each expression bin, supplemented with the distribution of the top 0.1% of correlations (within each expression bin). **Fig D: The background and signal components does not depend on expression level after spatial quantile normalization**. Data is 9 different GTEx tissues, all with 4 PCs removed. Like Fig C but after applying spatial quantile normalization. **Fig E: The impact of SpQN on transcription factor co-expression, all transcription factors**. Like Fig 7a. The percent increase in the number of edges (y-axis) identified after thresholding (x-axis) the correlation matrix, for edges involving transcription factors. **Fig F: The impact of SpQN on transcription factor co-expression, PPI transcription factors**. Like Fig 7b. The percent increase in the number of edges (y-axis) identified after thresholding (x-axis) the correlation matrix, for edges between genes with protein-protein interactions where one of the involved genes is a transcription factor. **Fig G: The relationship between the signal threshold (in percentage) and the expression level (before and after SpQN adjustment)**. We define the co-expression signal threshold (x-axis) as the top percentage of absolute correlation values (ranging between 0 and 3%). For a given signal threshold, we calculate the average expression level for each tissue (y-axis). We compare the expression levels before (blue) and after SpQN adjustment (pink). The average gene expression level for each tissue is shown by the dotted black line. **Fig H: The relationship between the percentage of co-expression signals and the expression bias**. Like Fig G, but where SVA was used to decide the number of PCs to be removed in the correlation matrix. **Fig I: The expression bias in graphical lasso network inference**. The expression levels of networks inferred by graphical lasso. Different values of the tuning parameter ($\rho$) results in different network sizes (x-axis) with higher values of the tuning parameter leading to smaller networks. The average gene expression for each tissue is shown by the dotted black line. **Fig J: IQR of Pearson correlations in each bin for 9 GTEx tissues (before and after SpQN adjustment)**. Bulk RNA-seq data from [15] from 9 tissues. Each tissue has a number of PCs removed based on the estimate from SVA, as suggested by [11]. **(a)** Background IQR for unadjusted (left smear) and SpQN-adjusted (right smear) gene-gene correlation distributions for all expression bins across 9 GTEx tissues. Color indicates expression level. **(b)** The relationship between sample size (x-axis) and IQR for correlations (y-axis) before and after adjustment. **Fig K: The background and signal components depend on expression level (before removing top PCs)**. Distributions of Pearson correlations for background genes (within each expression bin), supplemented with the distribution of the top 0.1% of correlations (within each expression bin). **Fig L: The effect of removing principal components on bias and variance of the background distribution**. **(a)** Average bias, defined as the average of the median of the 10 background distributions. **(b)** Average variance, defined as the average variance of the 10 background distributions. Red colored points have bias or variance exceeding the limits of the plot. **Fig M: The effect of removing principal components (PCs) on bias towards highly expressed genes**. As Fig 5d but for 9 tissues and two different approaches for removing PCs. **(a)** 4 PCs were removed from the correlation matrix. **(b)** SVA was used to estimate the number of PCs to remove (range: 10–30 PCs). **Fig N: ComBat does not remove the mean-correlation relationship**. Same raw data as Figs 2 and 3. We apply ComBat (with the date of nucleic acid isolation as batch variable) prior to constructing the expression matrix, instead of removing principal components. **(a)** Like Fig 3a, i.e. densities of the Pearson correlation between all genes within each of 10 expression bins (background) as well as the top 0.1. **(b)** Like Fig 2c, i.e. the

relationship between IQR of gene-gene correlation distribution and the lowest of the two expression bins associated with the submatrix. **Table A: Number of principal components to be removed, as estimated using SVA**.
(PDF)

## Author Contributions

**Conceptualization:** Yi Wang, Stephanie C. Hicks, Kasper D. Hansen.

**Data curation:** Yi Wang.

**Formal analysis:** Yi Wang.

**Funding acquisition:** Stephanie C. Hicks, Kasper D. Hansen.

**Investigation:** Yi Wang, Kasper D. Hansen.

**Methodology:** Yi Wang, Stephanie C. Hicks, Kasper D. Hansen.

**Project administration:** Kasper D. Hansen.

**Resources:** Kasper D. Hansen.

**Software:** Yi Wang, Kasper D. Hansen.

**Supervision:** Stephanie C. Hicks, Kasper D. Hansen.

**Validation:** Yi Wang, Stephanie C. Hicks, Kasper D. Hansen.

**Visualization:** Yi Wang, Kasper D. Hansen.

**Writing – original draft:** Yi Wang, Stephanie C. Hicks, Kasper D. Hansen.

**Writing – review & editing:** Yi Wang, Stephanie C. Hicks, Kasper D. Hansen.

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
