## [Decision Letter · Decision Letter 0]

21 Jul 2021

Dear Dr. Hansen,

Thank you very much for submitting your manuscript "Co-expression analysis is biased by a mean-correlation relationship" for consideration at PLOS Computational Biology.

As with all papers reviewed by the journal, your manuscript was reviewed by members of the editorial board and by several independent reviewers. In light of the reviews (below this email), we would like to invite the resubmission of a significantly-revised version that takes into account the reviewers' comments.

 One issue is that while the question for how to best exploit co-expression for biological discovery continues to be an important area of research, some indication of whether the approach is actually an improvement when it comes to biological inference. The reviewer acknowledges that this may be difficult to convincingly address, but at least needs to be acknowledged by the authors and discussed as a limitation. A second important point is the need for focus on comparisons to other potential approaches. The approach proposed is interesting and likely to help remove bias in gene correlation studies. There is a third issue that the reviewers do not mention but I believe is important to consider. In addition to mean expression level which you convincingly demonstrate is a driver of gene correlations, another driver of strong gene correlations in bulding co-expression networks is the statistical leverage of high differentials in expression occurring in genes across anatomic regions, or cells. Whereas the mean correlation will of course be raised by strong differentials, this mean by only be raised modestly overall. This is particularly important as most studies begin with a selection of higher differentially expression genes. It would be interesting to understand this effect and its relationship to or refinement of your model. Based on this feedback, I would like to invite you to submit a revised version of your manuscript addressing these considerations. 

We cannot make any decision about publication until we have seen the revised manuscript and your response to the reviewers' comments. Your revised manuscript is also likely to be sent to reviewers for further evaluation.

Sincerely,

Michael Hawrylycz

Guest Editor

PLOS Computational Biology

Ilya Ioshikhes

Deputy Editor

PLOS Computational Biology

Dear Dr. Hansen,

Your paper "Co-expression analysis is biased by a mean-correlation relationship" has been seen by three reviewers who indicate interest in your approach with several areas for clarification and follow up. One issue

is that while the question for how to best exploit co-expression for biological discovery continues to be an important area of research, some indication of whether the approach is actually an improvement when it comes to biological inference. The reviewer acknowledges that this may be difficult to convincingly address, but at least needs to be acknowledged by the authors and discussed as a limitation. A second important point is the need for focus on comparisons to other potential approaches. The approach proposed is interesting and likely to help remove bias in gene correlation studies. There is a third issue that the reviewers do not mention but I believe is important to consider. In addition to mean expression level which you convincingly demonstrate is a driver of gene correlations, another driver of strong gene correlations in bulding co-expression networks is the statistical leverage of high differentials in expression occurring in genes across anatomic regions, or cells. Whereas the mean correlation will of course be raised by strong differentials, this mean by only be raised modestly overall. This is particularly important as most studies begin with a selection of higher differentially expression genes. It would be interesting to understand this effect and its relationship to or refinement of your model. Based on this feedback, I would like to invite you to submit a revised version of your manuscript addressing these considerations.

Best regards,

Mike Hawrylycz

Guest Editor

PLoS Computational Biology

Reviewer's Responses to Questions

**Comments to the Authors:**

Reviewer #1: Overview:

The authors convincingly demonstrate a dependence between gene-gene correlations and gene expression. They propose a method to correct for this dependence and show that after applying this method, the dependence between correlation and expression is greatly reduced, perhaps even removed. As the authors note in the Discussion, this work may be applicable to much wider range of scenarios than those described in this work. My primary concerns focus on a lack of comparisons to other potential approaches. I have a few other comments and suggestions that I describe in detail below.

Major Comments:

1. It is difficult to fully appreciate the performance of the SpQN method without comparing it to another approach. While I appreciate that there aren't (to my knowledge) direct competitors, I would encourage the authors to consider whether there are other approaches that could serve as a comparison. For example, would applying a variance-stabilizing transformation largely eliminate the need for SpQN?

2. While this is becoming common practice in the field, regressing out the top N PCs likely removes substantial biological variation in addition to the technical variation they are assumed to represent. In the case of the GTEx data, for which RNA extraction and RNA sequencing dates are available, it would be interesting to see how the results change when one of these dates as a surrogate for batch and adjusting using ComBat.

3. This may be beyond the scope of the current manuscript, but I'd be curious to see the effect of considering sub-matrices of unequal size. For example, you could consider defining the size of the sub-matrices based on equal ranges of gene expression or equal within sub-matrix SDs.

Minor Comments:

1. I found Fig 2b and 5b unintuitive. Is there possibly a better way to show this information?

2. There appears to still be some widening of the background distributions and some shift in the signal distributions in Fig 5a, but it's hard to tell how much. I'd suggest adding vertical lines at -0.5, 0, and 0.5 to plots of this type.

3. I found the tissue-specific differences in background IQR (Fig 6a) interesting. Are these differences assumed to be purely technical?

4. The general rarity of negatively correlated gene pairs (except in a few tissues) also perhaps warrants further discussion.

5. There are quite a few typos throughout the paper that should be fixed.

Summary:

Overall, I think this is an important methodological advance in co-expression analysis and may open up other avenues of research. The paper would be strengthened by comparing to at least one (possibly naive) approach.

Reviewer #2: Wang et al. describe a computational method, “SpQN”, for normalizing coexpression data to remove the previously reported bias such data tends to have in favor of highly-expressed genes. The authors demonstrate the method can effectively remove this bias on bulk data from GTEx as well as single cell RNA-seq data. The method and the demonstrations of its efficacy are well-described and an R package is provided (which I did not test).

I believe this work has some merit, as the question for how to best exploit coexpression for biological discovery continues to be an important area of research. However, the authors give short shrift to the question of whether their approach is actually an improvement when it comes to biological inference. This may be difficult to convincingly address, but at least needs to be acknowledged by the authors and discussed as a limitation.

Specifically, while the authors have shown their method can reduce the mean-correlation bias, it seems possible it just introduces a new bias and needs to be evaluated. Will some truly biologically coexpressed gene pairs that are highly expressed be ignored after this normalization? Conversely, will some pairs of unrelated genes get inappropriately prioritized because they have low expression levels? As it stands, the manuscript primarily shows that the algorithm works at the task of removing the bias. But it is not clear that removing the bias is desirable. The real challenge (which I don’t expect to be met) is identifying lowly-expressed gene pairs that have been reproducibly and experimentally proven to be coexpressed or never coexpressed, and compare what happened to these pairs before and after the SpQN normalization.

The only biologically motivated validation reported here is that transcription factors have more coexpression after the normalization. However, this seems a guaranteed outcome of the normalization, assuming TFs tend to be lowly expressed, and it seems hardly worth saying that more and better are not synonymous. Whether having more coexpression for these genes is actually a good or bad thing is not assessed. Doing so might not be very easy. In bulk tissue, coexpression is likely driven largely by cellular compositional variation. This is potentially addressed by the PC removal procedure, but in any case, showing that discovery of true direct TF-target relations in enhanced would be a much more convincing demonstration.

An obvious other tack is to look at the PPIs of Luck et al, which is already used in the manuscript. But for many reasons this would also be a weak validation. For one thing there is no strong reason to expect protein levels to be strongly correlated with RNA generally, but it could be better than nothing.

In Crow 2016 and Farahbod 2019, the context was examining expression levels as an explanation for observed differences among the coexpression behavior of genes in different networks or different sets of genes within a network, not that the bias is an error in the construction of the networks that needs to be fixed. Presumably (but not demonstrated by the authors), SpQN would change the outcome of studies like Crow and Farahbod, but it is not clear whether the outcome would be desirable. For example, in Crow et al. it was observed that relative functional connectivity (as revealed by coexpression) of synaptic genes was partly explained by expression levels relative to other genes. Would removing the bias reduce that functional connectivity, or only obscure the fact that it is explained by expression levels? This seems worth discussion by the authors, if not explicit evaluation.

If stronger validation is not possible, this work should be positioned as a proposal of an interesting computational method for normalizing correlation matrices (whatever their source, not necessarily biological) with respect to an external variable (whatever it is), consistent with the authors’ suggestion at the end of the discussion that this is a general method. As it stands, I could not be enthusiastic about recommending the method for use at this stage of the research, because there is no evidence that it actually improves the outcome of coexpression analysis from a biology standpoint. But it may spark discussion and further work in the community.

Minor:

- The manuscript lacks a list of references. There is one in the preprint but it does not match the citations in the submitted manuscript, which I had to work around.

- Given the previous reports of this bias, the title and the way the work is presented seems to be a little bit of an overreach – both the title and the abstract makes it sound like the finding of the bias is original to this work. I’d suggest rewording to reflect that this work presents further documentation of the bias and a method for addressing it, not its discovery.

Reviewer #3: The review is uploaded as an attachment.

**Have the authors made all data and (if applicable) computational code underlying the findings in their manuscript fully available?**

Reviewer #1: Yes

Reviewer #2: **No: **There is a github repository with the code for the method, but I don't see the data used or code for analysis of the data presented in the manuscript. At least, it is not obvious as there is no documentation pointing to where they are.

Reviewer #3: **No: **The develop R package is publicly available in Github. I reckon the authors will make the analysis code available when paper is accepted.

PLOS authors have the option to publish the peer review history of their article (what does this mean?). If published, this will include your full peer review and any attached files.

Reviewer #1: **Yes: **Matthew N. McCall

Reviewer #2: No

Reviewer #3: No
---

## [Decision Letter · Decision Letter 1]

22 Feb 2022

Dear Mr. Hansen,

We are pleased to inform you that your manuscript 'Addressing the mean-correlation relationship in co-expression analysis' has been provisionally accepted for publication in PLOS Computational Biology.

Best regards,

Michael Hawrylycz

Guest Editor

PLOS Computational Biology

Ilya Ioshikhes

Deputy Editor

PLOS Computational Biology

Reviewer's Responses to Questions

**Comments to the Authors:**

Reviewer #1: I thank the authors for addressing all of my comments.

Reviewer #2: Thank you for the response and revisions. I appreciate the improvements made and am satisfied. I very much agree with the authors that the actual meaning of coexpression is often unclear and we are on the same page about how the proposed method should be considered by potential users.

**Have the authors made all data and (if applicable) computational code underlying the findings in their manuscript fully available?**

Reviewer #1: Yes

Reviewer #2: Yes

PLOS authors have the option to publish the peer review history of their article (what does this mean?). If published, this will include your full peer review and any attached files.

Reviewer #1: **Yes: **Matthew N. McCall

Reviewer #2: No

---

## [Editor Report · Acceptance letter]

16 Mar 2022

PCOMPBIOL-D-21-00859R1 

Addressing the mean-correlation relationship in co-expression analysis

Dear Dr Hansen,

I am pleased to inform you that your manuscript has been formally accepted for publication in PLOS Computational Biology. Your manuscript is now with our production department and you will be notified of the publication date in due course.

With kind regards,

Zsanett Szabo
